# ChatGPT for Fast Learning of Positive Energy District (PED): A Trial Testing and Comparison with Expert Discussion Results

**Xingxing Zhang \***[ID]**, Juveria Shah**[ID] **and Mengjie Han**[ID]

School of Information and Engineering, Dalarna University, 79188 Falun, Sweden; juh@du.se (J.S.); mea@du.se (M.H.)
\* Correspondence: xza@du.se; Tel.: +46-(0)-23-77-87-89

**Abstract:** Positive energy districts (PEDs) are urban areas which seek to take an integral approach to climate neutrality by including technological, spatial, regulatory, financial, legal, social, and economic perspectives. It is still a new concept and approach for many stakeholders. ChatGPT, a generative pre-trained transformer, is an advanced artificial intelligence (AI) chatbot based on a complex network structure and trained by the company OpenAI. It has the potential for the fast learning of PED. This paper reports a trial test in which ChatGPT is used to provide written formulations of PEDs within three frameworks: challenge, impact, and communication and dissemination. The results are compared with the formulations derived from over 80 PED experts who took part in a two-day workshop discussing many aspects of PED research and development. The proposed methodology involves querying ChatGPT with specific questions and recording its responses. Subsequently, expert opinions on the same questions are provided to ChatGPT, aiming to elicit a comparison between the two sources of information. This approach enables an evaluation of ChatGPT's answers in relation to the insights shared by domain experts. By juxtaposing the outputs, a comprehensive assessment can be made regarding the reliability, accuracy, and alignment of ChatGPT's responses with expert viewpoints. It is found that ChatGPT can be a useful tool for the rapid formulation of basic information about PEDs that could be used for its wider dissemination amongst the general public. The model is also noted as having a number of limitations, such as providing pre-set single answers, a sensitivity to the phrasing of questions, a tendency to repeat non-important (or general) information, and an inability to assess inputs negatively or provide diverse answers to context-based questions. Its answers were not always based on up-to-date information. Other limitations and some of the ethical–social issues related to the use of ChatGPT are also discussed. This study not only validated the possibility of using ChatGPT to rapid study PEDs but also trained ChatGPT by feeding back the experts' discussion into the tool. It is recommended that ChatGPT can be involved in real-time PED meetings or workshops so that it can be trained both iteratively and dynamically.

**Keywords:** ChatGPT; PED; challenge; impact; communication and dissemination





## 1. Introduction

A positive energy district (PED) is an area of urban development designed and managed so that it generates a high share of locally generated renewable energy. Neighborhoods and other shared urban spaces actively work towards the generation of an annual energy surplus, thus creating sustainable, self-sufficient communities that are, to a large extent, energy-neutral with a low carbon footprint [1]. A PED can be created through the integration of various energy-efficient technologies and renewable energy sources such as solar panels, geo-thermals, and other energy-saving systems. The PED is an important mechanism for achieving climate neutral cities. Climate neutrality refers to the goal of reducing greenhouse gas emissions to zero and balancing any remaining emissions with equivalent carbon credits or other activities that remove carbon from the atmosphere [1,2]. A PED can play a significant role in achieving this goal by reducing energy consumption

and emissions, producing renewable energy, and providing a model for sustainable development that can be replicated in other parts of the city and beyond. PEDs can therefore also help to raise awareness about the importance of sustainability and encourage people to adopt more sustainable behavior [3].

There are many existing studies about PEDs from concept design and planning to implementation and operation. In terms of design and planning, studies have explored the role of urban planning and architectural design in shaping the energy performance of positive energy districts. Researchers have found that the placement and orientation of buildings, the choice of materials and technologies, and the design of public spaces can all play a significant role in shaping the energy performance of a PED [4]. In terms of implementation and operation, studies have evaluated the energy performance of completed PEDs and found that these developments can achieve significant energy savings through the integration of renewable energy sources, energy-efficient technologies, and other sustainability measures [5]. However, several challenges exist in the design and planning of the new urban structures that dominate many PEDs. These include the need for investment in new infrastructure, changes in building codes and regulations, and the need for coordinated planning and management [6]. There are also challenges associated with the operation and maintenance of PEDs, including the need for ongoing monitoring and management, the need for coordinated energy management systems, and the need for long-term financing and investment [7,8]. However, navigating the different stages of design, planning, implementation, and operation of PEDs presents unique challenges that need to be addressed for successful outcomes:

*Design Stage*:

- Integration of energy systems: Designing and integrating various renewable energy sources, energy storage systems, and energy-efficient technologies into the district's infrastructure.
- Urban form and spatial planning: Balancing the energy requirements with the design of buildings, transportation systems, and public spaces to optimize energy efficiency and create a livable environment.

*Planning Stage:*

- Policy and regulatory framework: Developing supportive policies and regulations that facilitate the implementation of PEDs, such as zoning regulations, building codes, and renewable energy incentives.
- Stakeholder engagement: Engaging and involving various stakeholders, including residents, businesses, local authorities, and utility companies, in the planning process to ensure their input and support.

*Implementation Stage:*

- Funding and financing: Securing sufficient financial resources for the construction and installation of the necessary infrastructure, which often requires innovative financing models and collaboration between public and private entities.
- Project management: Ensuring effective coordination and management of multiple stakeholders, contractors, and suppliers to ensure timely and efficient implementation.

*Operation Stage:*

- Energy management and optimization: Monitoring and optimizing energy generation, consumption, and distribution within the district to maintain positive energy balance and minimize waste.
- User behavior and engagement: Encouraging residents and businesses to adopt sustainable behaviors and engage in energy-saving practices to maximize the benefits of the PEDs.
- Data management and monitoring: Establishing effective systems for collecting, analyzing, and utilizing data on energy performance and user behavior to continuously improve the operation and efficiency of the PEDs [9].

Existing urban neighborhoods, particularly in older and culturally protected parts of a city, can be seen as another key barrier for PED mainstreaming [10,11]. Overall, existing studies suggest that PEDs have the potential to balance and support the energy grid through local energy generation, the reduction of energy consumption and greenhouse gas emissions, and the promotion of sustainability in urban areas. However, the solutions to these different challenges require the integrated effort of all stakeholders, where the effective **communication and dissemination** are essential to ensure clear and comprehensive knowledge transfer from experts to the general public in the urban transition, including citizens, utility companies, real estate developers, banks, and other stakeholders. This entails the need for transparent and accessible information sharing to facilitate a widespread understanding between public officers and practitioners.

Although PEDs have been a subject of considerable scholarly research, the average citizen still does not know enough about these urban spaces to support their development. **This is a critical challenge in PED dissemination** for several reasons, including: (1) Lack of awareness: The concept of a PED is still relatively new and has not received widespread media attention or public education. As a result, many people may not have heard about it and do not fully understand its purpose; (2) Technical jargon: The topic of energy production and sustainability can be complex and technical, and the use of technical language and specialized terms can make it difficult for the average person to understand; (3) Disconnect from daily life: For some people, the idea of the community-based energy management system that a PED requires may seem abstract and removed from their daily lives, making it difficult for them to grasp the significance of the concept; (4) Misconceptions: There may be misconceptions or myths about the concept of a PED that prevent people from fully understanding it. Indeed, more education and outreach efforts are needed to raise awareness and understanding of PEDs and their importance in promoting sustainability and reducing the impact of climate change [12].

To overcome these challenges and promote the greater public understanding of PEDs artificial intelligence (AI), which has shown promise in other domains, has the potential to automate experience gaining and knowledge exploration. Among the potential AI techniques currently available, language models have been an active research topic in the field of natural language processing (NLP) in recent years, and their progress has been remarkable. NLP is a subfield of artificial intelligence that focuses on enabling computers to understand, interpret, and generate human language. It involves developing algorithms and models that can analyze, process, and produce natural language text or speech. NLP is used in a wide range of applications, such as chatbots, language translation, sentiment analysis, speech recognition, and text summarization [13].

A language model is a type of AI that is trained to understand human language and generate text that mimics natural language. It uses statistical techniques to predict the likelihood of words or phrases occurring in a given context, and can be used for tasks such as language translation, text summarization, and speech recognition [14]. Although applications of language models to different dimensions of PEDs are still limited, they have demonstrated the feasibility and advantages for their further utilization and exploration. For example, by using a language model, a standard metadata representation would increase the cost-effectiveness of deploying smart buildings in PEDs [15]. Such a model could automate the extraction of regulatory information by using POS-tagged building code data [16]. Regarding the use of renewable energy and energy markets, social media data could fine-tune the model to examine public attitudes [17,18].

A chatbot is a computer program designed to simulate conversation with human users, especially over the internet. Like a smart entity, it uses AI and NLP to understand and respond to user queries or commands [19]. Modern chatbots can be integrated into various platforms such as messaging apps, websites, and mobile apps to provide quick and efficient customer service, information, or support by utilizing various computational approaches [20,21]. ELIZA is considered to be one of the earliest chatbot models, and it was developed in the mid-1960s at MIT [22]. ELIZA used pattern recognition and simple

rules to mimic a psychotherapist. ELIZA operated by inputting the words that users typed into a computer and then matching them against a set of predetermined responses. It was not designed to provide genuine therapy but rather to demonstrate the limitations of communication between humans and machines. In the 1970s, PARRY, a chatbot designed to mimic a person with paranoid schizophrenia, was developed by Kenneth Colby at Stanford University [23,24]. PARRY used a rule-based approach to generate responses, where a framework of assumptions, attributions, and "emotional responses" were activated by adjusting the weights assigned to the user's verbal inputs. PARRY was groundbreaking, as it demonstrated the potential of computer programs to simulate complex human behaviors and mental states. In the 1980s and 1990s, chatbots became more sophisticated with the use of NLP techniques. Programs such as Racter [25] and Jabberwacky [26] used NLP to generate responses based on statistical models of language. Racter was created to generate surreal and nonsensical responses that were intended to be humorous. It used a language generation program that combined a Markov chain algorithm with a template-based approach to produce responses. Jabberwacky's development has paved the way for other technological progressions, and its website has been used by researchers for academic purposes. In recent years, chatbots have become increasingly popular with the rise of AI and machine learning. Chatbots now use advanced algorithms and machine learning techniques to understand natural language and generate responses, for example, IBM Watson [27], ALICE [28] and voice- or speech-based assistants [29,30].

As the cornerstone of language models, the transformer architecture adopts self-attention mechanisms to process sequences of inputs and has become the basis for many state-of-the-art models in NLP [31]. For example, by using over 175 billion parameters, Generative Pre-Trained Transformer 3 (GPT-3) is one of the most extensive language models and has exhibited remarkable success in various NLP tasks, such as language generation, translation, and summarization [32]. It also has the potential to revolutionize certain aspects of scientific writing and information dissemination. Another pre-trained and transformer-based language model, BERT, also achieves state-of-the-art performance on a broad spectrum of NLP tasks, such as sentiment analysis, named entity recognition, and question answering by utilizing bidirectional attention mechanisms and masked language modeling [33]. To overcome the resulting computational challenges, researchers have developed efficient transformer models such as the reformer, which improves computational efficiency while maintaining performance [34]. The reformer uses a novel mechanism called locality-sensitive hashing to achieve logarithmic time and linear memory complexity. T5 is a text-to-text model that showcases the versatility of the transformer architecture. It demonstrates that the transformer architecture can be effectively applied to a wide range of NLP tasks [35].

Fine-tuning is a crucial component in the development of language models, which uses a pre-trained language model and adapts it to task-specific training using customized data. For example, ULMFiT [36] allows pre-trained models such as ELMo [37] to be adapted to specific NLP tasks by gradually unfreezing and fine-tuning the model's layers. This method has been demonstrated to be a successful approach in tasks such as sentiment analysis and text classification. Variants of pre-trained models, including RoBERTa [38] and XLNet [39], have also been proposed to address limitations in the original BERT model, while ALBERT [40] offers a lightweight version of BERT using parameter sharing and factorized embedding matrices to enable faster training and improved memory efficiency.

GPT refers to a family of natural language processing models that use deep learning techniques, specifically transformers, to generate human-like responses to text inputs. GPT models are pre-trained on massive amounts of data, allowing them to generate coherent and contextually relevant responses to a wide range of prompts [41,42]. OpenAI has been a prominent contributor to NLP and has been at the forefront of the development of large-scale GPT language models. In 2018, OpenAI introduced the first of a series of GPT language models, GPT-1 [41]. This was followed by the introduction of GPT-2 [43], GPT-3 [32], and GPT-4 [44], which have been trained on even larger amounts of text

data and have demonstrated impressive levels of accuracy and fluency in various tasks. Their chatbot-focused GPT-3 variant, known as ChatGPT, has demonstrated a utility in applications beyond NLP tasks. ChatGPT has been trained on a large dataset of human conversations and can generate contextually relevant and natural-sounding responses to conversational prompts.

As research in the field of chatbots continues to evolve, it is likely that more impressive results and a wide range of applications will emerge in the future. On the other hand, there are essential challenges in dissemination of PED concepts and results, which are necessary to overcome. Thus, there is a strong need to examine the possibilities of using ChatGPT, as a representative language tool, for fast and widespread dissemination of PEDs. This study represents a tentative effort in testing this idea by conducting a comparative analysis between the results obtained from ChatGPT and the expert opinions derived from practical PED workshops.

Section 2 explains the research methodology, and Section 3 outlines the results from ChatGPT and the practical PED workshops. Section 4 presents a discussion of this trial testing. Conclusions and closing remarks are laid out in Section 5.

## 2. Research Methodology

This study focused on three key topics of PEDs, in terms of challenges, impact and communication and dissemination. The discussion results of the participating experts were used to compare and validate the results generated from ChatGPT. Moreover, the experts' opinions were further fed back into the ChatGPT, and an iterative comparison was then carried out. Figure 1 illustrates the basic research principle.

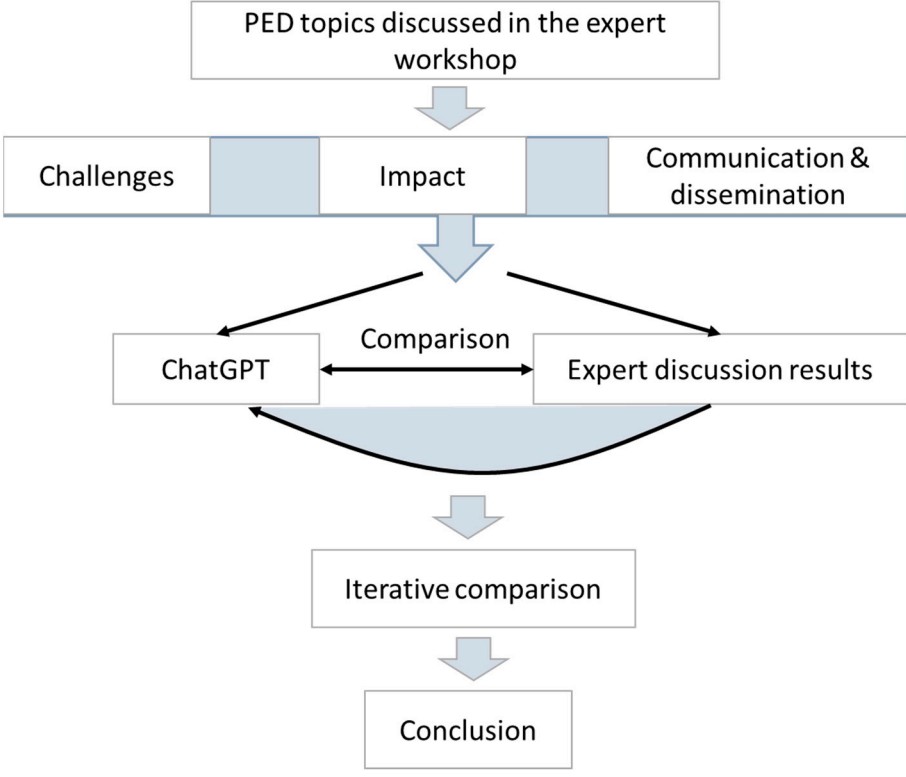

**Figure 1.** Flow chart of the general research method.

### 2.1. ChatGPT

A model of reinforcement learning from human feedback (RLHF) [45,46] was trained by supervised fine-tuning, where human AI trainers provided conversations and they played both sides—the user and an AI assistant. The trainers had access to model-written suggestions to help them compose their responses. This new dialogue dataset was then

integrated with the InstructGPT dataset, which had been transformed into a dialogue format. ChatGPT is built upon InstructGPT and uses a transformer architecture, which allows it to capture long-range dependencies in text and generate coherent responses that are contextually relevant. Our results in this paper are based on GPT-3.5, which finished training in early 2022 on an Azure AI supercomputing infrastructure. OpenAI employed a group of approximately 40 contractors who were attuned to the preferences of various demographic groups to generate demonstration and comparison data. Thus, the range of tasks covered by the inputs is significantly wider [47]. Specifically, as indicated in Figure 2, three steps were performed to train the model.

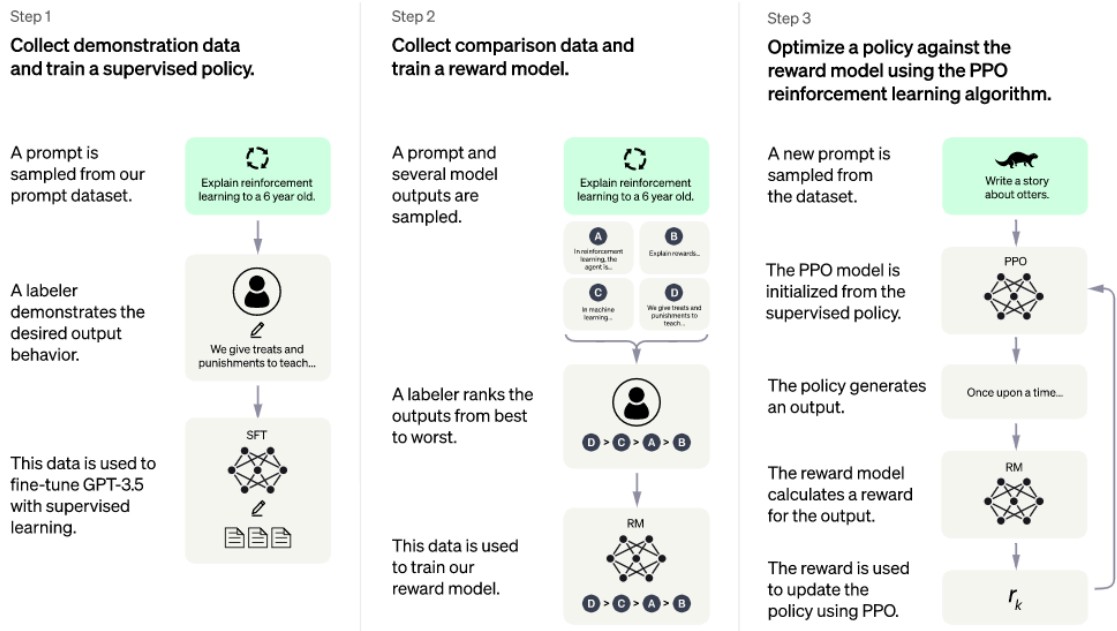

**Figure 2.** Working principles of ChatGPT [47].

*Step 1 Supervised fine-tuning (SFT)—GPT-3* [32] is trained on a broad distribution and massive quantity of internet data such that it can be adjusted to suit a diverse set of downstream tasks. It is then fine-tuned on the labeler demonstrations using supervised learning.

*Step 2 Reward model (RM)—An RM* [46] is trained to predict the human-preferred output on a dataset of comparisons between two model outputs on the same input. The variation in rewards signifies the logarithmic odds of a human evaluator favoring one response over the other. This reward signal is then used to update the language model's parameters, encouraging it to generate better responses that are more likely to satisfy the given task or objective. It provides a way to guide the language model towards generating responses that are not only contextually relevant but also satisfy a specific objective, such as being informative, concise, or engaging.

*Step 3 Reinforcement learning—The SFT model* is further trained using proximal policy optimization (PPO) [48]. After receiving the prompt and generating a response, based on the reward model, the system produces a reward and terminates the episode.

However, according to the limits [47], ChatGPT sometimes writes plausible sounding but incorrect or nonsensical answers. Therefore, it is necessary to tackle this challenge by providing sources of truth from practice, such as expert discussion.

ChatGPT can answer questions, provide suggestions, and even generate options for consideration, which can largely assist humans in making decisions. It can identify potential improvements to alert logic and support decision and assist experts in formulating their own suggestions for improvement based on complex information [49,50]. Kashyap discussed the launch of ChatGPT by OpenAI and its potential as a step towards creating AI [51]. The author suggested including the principle of inclusion in AI systems and

discussed pedagogical possibilities and concerns about job redundancy. He emphasized the incorporation of the inclusion principle as the fundamental prerequisite for all intelligent systems, including ChatGPT. George and George argued that the objective of ChatGPT is to reduce misunderstandings between people and machines during interactions [52]. OpenAI hopes that ChatGPT's advanced NLP techniques will lead to better conversational AI agents that can perform complex tasks while making it easy for people to talk to them. They concluded that ChatGPT is a significant step forward in improving the way computers and people communicate and could have various applications. Aydin and Karaarslan conducted a literature review using only ChatGPT, which culminated in encouraging results [53]. They attempted to demonstrate how AI will speed up the collection and expression of knowledge and claimed that future academic publication procedures will involve less human labor, allowing academics to concentrate on their research. Aljanabi discusses how with the integration of other AI technologies and the potential for enhanced personalization and customization, the future of ChatGPT holds numerous exciting opportunities to positively impact our lives. Advancements in language model performance, driven by improved training algorithms and expanded datasets, will be pivotal in shaping ChatGPT's evolution, opening doors to novel applications in healthcare, finance, and other domains reliant on sophisticated data analysis and understanding [54]. Mijwil et al. used ChatGPT to write a paper about the role of cybersecurity in the protection of medical information and concluded that ChatGPT showcases its proficiency in producing scholarly and sophisticated content, presenting the potential for authors to create customized sections through interactive chat. They anticipate AI to increasingly aid researchers in crafting scientific articles and assume a pivotal role in advancing scientific research [55].

### 2.2. Experts Workshop

A workshop on PEDs with eighty-eight experts from different fields and different European countries has been held recently. It needs to be stated that the paper only aims to demonstrate the usefulness of integrating work with ChatGPT in dissemination and communication issues at the concrete example of the PED concept. Therefore, the content (by both ChatGPT and the experts) is not the focus but serves as a demonstration/comparison. Regarding the experts' statements, there has not yet been an agreed summary of the workshop outcome apart from the individual post. So, for this purpose, we decided to completely anonymize the workshop setting.

Three main topics were discussed during the workshop: the challenges of PED development; the impact of PEDs; and the communication and dissemination of knowledge of PEDs. Within each category, a number of sub-issues were addressed as shown in Figure 3. These are outlined in Figure 1 and were then used as the input questions for ChatGPT:

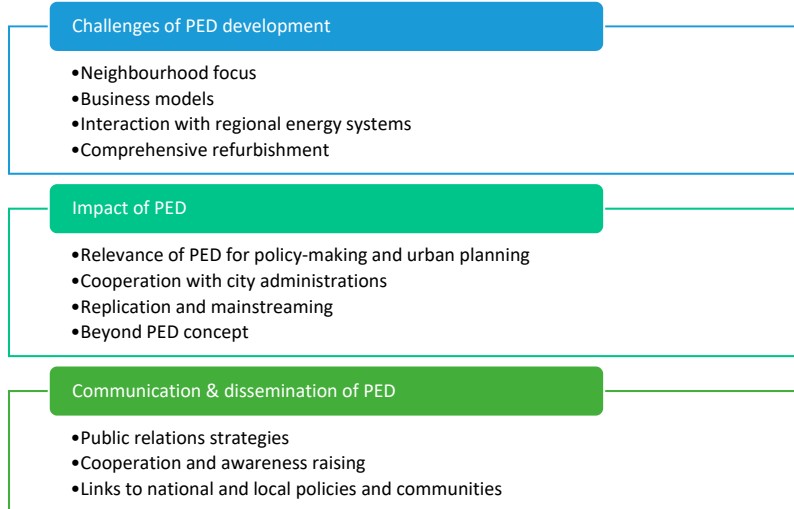

**Figure 3.** Three main topics for experts' workshop.

## 3. Results from Both ChatGPT and Practical PED Workshops

This section provides the comparison results from both ChatGPT and practical PED workshops. The expert's opinions in each topic were further fed back to ChatGPT to train the model iteratively. It needs to be mentioned that there were diverse opinions from the experts on different topics during the workshop. The observations and suggestions are only summaries by the authors.

*3.1. Challenges of PED Development*

3.1.1. Neighborhood Focus

*Q: What are some neighborhood related problems that will challenge PEDs?*

**ChatGPT**: PEDs face several neighborhood-level challenges that can hinder their development and implementation, including:

- Resistance to Change: Residents may resist changes to their community, especially if they are skeptical about the benefits of a PED or fear that it will lead to negative impacts such as increased costs or reduced quality of life.
- Lack of Awareness: Residents may be unaware of the benefits of a PED or lack an understanding of how it works, making it difficult to engage them and build support.
- Social and Economic Disparities: PEDs can face challenges in lower-income or economically disadvantaged neighborhoods, where residents may have limited resources to participate or may be skeptical of changes that they see as benefiting wealthier communities.
- Technical Challenges: Developing the necessary infrastructure and technology to support a PED can be complex, especially in older or densely populated neighborhoods with limited space and resources.
- Aging Infrastructure: Neighborhoods with aging infrastructure may face challenges in upgrading their energy systems to support a PED, especially if they lack the resources or technical capacity to make the necessary upgrades.
- Community Management: Ensuring effective community management and governance can be challenging, especially in larger or more complex communities with multiple stakeholders.

**Experts**: the observations and suggestions made by experts include:

- Localising energy system.
- Building social connection, shared identities, monitoring changing behaviours, fostering participation.
- Identifying stakeholders and financing source.
- Providing incentives and developing financial models.
- Addressing energy poverty.
- PED and energy communities mixed use/balance.

**Iterative training:** these observations and suggestions highlight the importance of collaboration, community engagement, and innovative financing and business models in creating successful PEDs.

- Localizing energy systems is essential for reducing energy consumption and improving energy efficiency. By creating decentralized energy systems that are tailored to local conditions, PEDs can maximize their impact and reduce the need for large, centralized energy infrastructure.
- Building social connections, monitoring changing behaviors, and fostering participation are important for engaging residents and creating a sense of community around the PED. By involving residents in the development and operation of the PED, it becomes more likely that they will adopt sustainable energy practices and behaviors.
- Identifying stakeholders and financing sources is critical for ensuring that the necessary resources are available to implement a PED. By engaging with key stakeholders and securing sufficient financing, PEDs can be developed and operated effectively.
- Providing incentives and developing financial models can help overcome technical and financial challenges and ensure that a PED is sustainable over the long term. By

creating incentives for residents to participate in the PED and developing financial models that are tailored to local conditions, PEDs can maximize their impact and ensure their long-term success.

### 3.1.2. Business Models

*Q: What are some business model challenges for PEDs?*

**ChatGPT:** PEDs can face several business model challenges, including:

- Financing: Finding the necessary financing to develop and implement a PED can be challenging, especially if the costs of implementing new energy infrastructure and technology are high.
- Cost-Effectiveness: Ensuring that the cost of implementing a PED is affordable and cost-effective for residents can be challenging, especially in lower-income neighborhoods where residents may have limited resources.
- Revenue Generation: Developing a sustainable revenue model that supports the ongoing operation and maintenance of a PED can be challenging, especially if the revenue generated by the PED is insufficient to cover its costs.
- Stakeholder Cooperation: Ensuring effective cooperation and collaboration among all stakeholders, including residents, energy companies, local authorities, and financial institutions, can be challenging, especially if there are competing interests or conflicting goals.
- Regulation: Navigating the complex regulatory environment, including energy regulations, building codes, and zoning laws, can be challenging, especially if the regulations are unclear or inconsistent.
- Technical Capacity: Ensuring that all stakeholders have the necessary technical knowledge and skills to participate in the PED can be challenging, especially if they lack access to training and support.

**Experts**: some observations and suggestions made by experts are:

- Support from municipalities for financing PEDs.
- Establishing partnership between governments and citizens, while providing information about co-benefits.
- Developing new business models for decentralization and support existing energy providers to overcome difficulties.
- Capacity-building in local administrations needed.
- Establishing energy cooperatives.

**Iterative training:** these suggestions highlight the importance of collaboration, transparency, and innovation in creating successful business models for PEDs.

- Support from municipalities for financing PEDs is critical to ensuring the necessary resources are available to implement a PED effectively.
- Establishing partnerships between governments and citizens and providing information about co-benefits is essential for engaging residents and building support for a PED.
- Developing new business models for decentralization and supporting existing energy providers can help overcome technical and financial challenges and ensure that a PED is sustainable over the long term.

### 3.1.3. Interaction with Regional Energy Systems

*Q: What are the challenges for interaction with regional energy systems for PEDs?*

**ChatGPT:** PEDs are designed to promote decentralized energy generation and consumption, and as a result, they face several challenges in their interaction with regional energy systems:

- Integration with existing energy infrastructure: Integrating PEDs with existing energy infrastructure, such as distribution networks and grids, can be challenging, as it requires coordination between different stakeholders and the development of new technical solutions.

- Balancing energy supply and demand: PEDs rely on decentralized energy sources, such as solar and wind, which are subject to fluctuations in availability. Balancing energy supply and demand in PEDs can be a challenge, especially in regions with high levels of renewable energy generation.
- Energy storage: Energy storage is critical for ensuring that energy generated by PEDs can be used when it is needed, rather than being lost when supply exceeds demand. However, energy storage solutions are still expensive, and their implementation in PEDs can be challenging.
- Regulating the energy market: The energy market is regulated by national and regional authorities, and PEDs may face challenges in their interaction with these regulations, especially if they seek to create new business models for energy production and consumption.
- Managing energy demand: PEDs rely on energy-efficient technologies and behaviors to reduce energy consumption, but managing demand can be challenging, especially if residents are not fully engaged in the PED.
- These challenges highlight the need for ongoing collaboration between PEDs and regional energy systems, as well as the importance of innovation and technology to overcome these challenges and ensure that PEDs are able to achieve their goals.

**Experts:** some observations and suggestions made by experts are:

- Achieving zero $CO_2$ emission goal and positive energy balance.
- Establishing PED as a vision and transformation as a process.
- Capacity limits of grids → need for flexibility solutions.
- Interaction between PEDs.
- Implement a "system of systems" in the energy grid.

**Iterative training:** these observations and suggestions reflect the key challenges for PEDs and their regional energy system:

- Achieving a zero $CO_2$ emission goal and a positive energy balance is a key objective for PEDs, as it requires the integration of energy-efficient technologies and behaviors with decentralized energy generation and consumption. This requires investment and collaboration between various stakeholders, including citizens, businesses, and governments at local and regional levels.
- Establishing PEDs as a vision and transformation as a process is also important. PEDs are not just about technology, but also about social and cultural change of local and regional areas. Building a vision for the PED, and engaging residents and businesses in the transformation process, is essential for ensuring that the PEDs are able to achieve their regional goals and meet the needs of all stakeholders.

### 3.1.4. Comprehensive Refurbishment

*Q: What are the challenges for Comprehensive refurbishment for PEDs?*
**ChatGPT:** Comprehensive refurbishment of buildings in PEDs presents several challenges, including:

- High upfront costs: Refurbishing buildings to meet high energy efficiency standards requires significant investment, which can be a barrier for residents and businesses.
- Technical expertise: Comprehensive refurbishment requires technical expertise in building design, construction, and energy systems, which can be a challenge in regions with limited skilled labor.
- Financing: Access to financing can be a challenge, especially for residents and small businesses, who may not have the resources to invest in refurbishment.
- Resistance to change: Some residents and businesses may resist change, especially if they are not fully engaged in the PED, or if they have concerns about the cost or impact of refurbishment.
- Legal and regulatory barriers: Refurbishment may also face legal and regulatory barriers, such as building codes and regulations, which can limit the scope of refurbishment or delay the implementation of refurbishment projects.

These challenges highlight the importance of comprehensive planning and engagement with all stakeholders, as well as the need for innovative financing solutions, such as public-private partnerships, to support comprehensive refurbishment in PEDs.

**Experts:** some observations and suggestions made by experts are:

- Visible improvement in quality of life.
- Clear and long -term vision for urban/regional development, while working with the specific local context.
- Maintaining architectural heritage and providing master planning.
- Introducing energy academy for residents.
- District heating should be "non-profit": run by municipalities and/or ownership by consumers.
- Investment needs to be related to business case.

**Iterative training:** these observations and suggestions reflect some of the key considerations for comprehensive refurbishment in PEDs.

- Visible improvement in quality of life is an important goal of comprehensive refurbishment, as it can help to engage residents and businesses in the PED and to ensure that the benefits of refurbishment are widely understood and appreciated.
- A clear and long-term vision for urban and regional development is also important, as it provides a framework for refurbishment that is aligned with the wider goals and aspirations of the community.
- Maintaining architectural heritage and providing master planning can help to ensure that refurbishment is integrated into the fabric of the community and that it supports the preservation of important cultural and historic assets.
- An energy academy for residents can provide residents with the skills and knowledge they need to participate in the PED, and to take an active role in ensuring that the PED meets the needs of all stakeholders. This can help to foster a sense of community ownership and engagement in the PED.

*3.2. Impact of PEDs*

3.2.1. Impact of PEDs on the Policy-Making and Urban Planning "Real World"?

*Q: How to ensure relevance of the PED projects in the policy-making and urban planning "real world"?*

**ChatGPT:** To ensure the relevance of PEDs in the policy-making and urban planning "real world", the following steps can be taken:

- Stakeholder engagement: Engage a diverse range of stakeholders, including government agencies, local communities, energy utilities, and businesses to ensure that PED projects align with their needs and priorities.
- Data-driven decision making: Use data and evidence to inform policy and planning decisions, to ensure that PED projects are informed by the best available information and are evidence-based.
- Integration with existing policies and plans: Ensure that PED projects are integrated with existing policies and plans for energy, sustainability, and urban development, to ensure that they are aligned with broader efforts to achieve sustainable and low-carbon communities.
- Collaboration with other initiatives: Collaborate with other initiatives, such as Smart Cities and Renewable Energy projects, to maximize synergies and avoid duplication of effort.
- Monitoring and evaluation: Regularly monitor and evaluate PED projects to track progress, identify challenges, and refine the approach to ensure that PED projects continue to meet the evolving needs of communities.

By following these steps, PED projects can be better aligned with the needs of communities and be better positioned to contribute to the development of sustainable, low-carbon, and resilient communities.

**Experts:** some observations and suggestions made by experts are:

- Capacity-building in cities (trainings, "train the trainer").
- Improved data collection and data accessibility, digital twins.

**Iterative training:** these observations and suggestions made by experts regarding capacity-building and improved data collection and accessibility are relevant and important for ensuring the relevance of PED projects in policy-making and urban planning.

- Capacity-building can help city officials and other stakeholders better understand the benefits of PEDs and how they can be implemented in real-world contexts.
- Improved data collection and accessibility can help to ensure that the projects are evidence-based and can be effectively monitored and evaluated over time.
- Digital twins can be a useful tool for urban planners to simulate and test the impacts of PEDs on the urban environment, helping to inform policy decisions and community engagement efforts.

### 3.2.2. Cooperation with City Administrations

*Q: What are the impacts of Cooperation with city administrations for PEDs*

**ChatGPT:** Cooperation with city administrations is critical for the success of PEDs. The following are some of the impacts of cooperation with city administrations for PEDs:

- Access to resources: City administrations can provide access to resources such as funding, staff, and data that can support the implementation of PED projects.
- Alignment with policy: By working with city administrators, PEDs can align with existing policies and regulations related to energy, sustainability, and urban development, which can improve the feasibility and impact of PED projects.
- Improved engagement with communities: Working with city administrators can help to build trust and increase engagement with local communities, which is essential for the success of PED projects.
- Scalability: Cooperation with city administrators can help to scale up PED projects to a larger number of communities, increasing their impact and sustainability.
- Improved planning and coordination: City administrators have a wealth of experience in planning and coordinating large-scale projects, which can support the development of effective PED projects.
- Enhanced visibility: Cooperation with city administrators can increase the visibility of PED projects, increasing awareness and understanding of their benefits and encouraging wider adoption.

In summary, cooperation with city administrators is essential for the success of PED projects and can help to enhance the impact and sustainability of PED initiatives.

**Experts:** some observations and suggestions made by experts are:

- Re-consideration of the PED narrative, better link to cities' strategies
- Stronger involvement of cities in research & innovation PED projects

**Iterative training:** these observations and suggestions made by experts about the impacts of cooperation with city administrations for PEDs are valid and important.

- Re-considering the PED narrative to better align with cities' strategies can help ensure that PED projects are relevant and responsive to local needs and priorities.
- Stronger involvement of cities in research and innovation PED projects can help build stronger partnerships between local authorities, researchers, and other stakeholders, leading to more effective and sustainable projects.

### 3.2.3. The Value of Practice and the Issue of Replication and Mainstreaming

*Q: What are the impact value of (best) practice and the issue of replication and mainstreaming for PEDs?*

**ChatGPT:** The impact value of best practices and the issue of replication and mainstreaming are critical components of PEDs. The following are some of the impacts of best practices and replication/mainstreaming for PEDs:

- Increased impact: Adopting best practices in PED projects can increase their impact, helping to achieve energy efficiency, reduce greenhouse gas emissions, and improve the quality of life in communities.
- Scalability: Replication of successful PED projects can help to scale up their impact, leading to more widespread adoption and greater sustainability.
- Mainstreaming: Mainstreaming PEDs into wider policy and urban planning frameworks can help to embed the principles of PEDs into long-term strategies, increasing their sustainability and impact.
- Knowledge sharing: By sharing best practices and replicating successful PED projects, knowledge and expertise can be shared across communities, leading to more effective and efficient implementation of PED initiatives.
- Cost-effectiveness: Replicating successful PED projects can lead to more cost-effective implementation, as lessons learned from previous initiatives can be applied to new projects.
- Increased stakeholder engagement: Replicating and mainstreaming PED initiatives can increase engagement with stakeholders, including government, businesses, and communities, which is essential for their success.

**Experts:** some observations and suggestions made by experts are:

- Continuous work with the projects, ensure pick-up after project ends towards implementation (funding chain?)
- Applied communication strategies—key results, benefits and process guidance for city administrations.

**Iterative training:** these observations and suggestions made by experts about the impact value of best practices and the issue of replication and mainstreaming for PEDs are useful.

- Continuous work with the projects is necessary to ensure that the projects are implemented after the funding ends. It is important to create an applied communication strategy that focuses on the key results, benefits, and process guidance for city administrations. This will help to raise awareness of the potential benefits of PEDs and encourage the replication and mainstreaming of successful practices.
- Additionally, exploring funding opportunities and establishing a funding chain can help to ensure that PEDs are implemented and maintained over the long-term.

### 3.2.4. Beyond PED Concept

*Q: What will be impact of linking PEDs to digitalization and circularity?*
**ChatGPT**: Linking PEDs to digitalization can have several impacts, such as:

- Improved energy management: By incorporating digital technologies, PEDs can more effectively manage energy generation, distribution, and consumption, leading to improved energy efficiency and reduced waste.
- Increased transparency: Linking PEDs to digitalization can provide greater visibility into energy usage and enable better monitoring of energy production and consumption, leading to increased transparency and accountability.
- Enhanced engagement: Digital technologies can provide new ways for residents and communities to engage with PEDs and participate in decision-making, promoting greater community involvement and ownership.
- Improved decision-making: The integration of digital technologies into PEDs can provide more robust data and analytics to support informed decision-making, leading to more effective energy management and planning.
- Support for innovation: Linking PEDs to digitalization can facilitate the development of new energy solutions and support innovation in the energy sector.
- Reduced waste: Circular principles aim to reduce waste and make the most efficient use of resources. By linking PEDs to circularity, energy systems can be designed to minimize waste and promote the reuse and recycling of resources.

- Increased sustainability: Linking PEDs to circularity can support the transition to a more sustainable energy future by promoting the efficient use of resources and reducing the negative impacts of energy production and consumption.
- Improved resource management: By incorporating circular principles into PEDs, energy systems can be designed to make the most efficient use of resources, reducing the need for new energy resources and helping to conserve existing ones.
- Support for local communities: Circular approaches can promote the development of local energy systems and support the growth of local communities, creating new economic opportunities and promoting greater community involvement and ownership.
- Increased resilience: By reducing dependence on non-renewable resources, linking PEDs to circularity can increase the resilience of energy systems and support their ability to adapt to changing conditions and challenges.

**Experts:** some opinions from experts are listed as below:

- Continuality
  - Work with learnings.
  - Set up reflection groups.
  - Connection to EU projects.
  - Train the trainer.
  - Task force, such as interdepartmental.
  - Involve the city/municipality, raise awareness, built capacity.
  - Exchange between the projects, bring results to administration, see common learning lessons.

- Data
  - Improve data collection.
  - Data accessibility and expertise should be increased.
  - EU survey on national data collection for energy transition.
  - Visualization of PED based on Data.
  - Digital twins.
  - Can data models shape urban reality, better understanding needed?
  - Better quality of research & innovation projects.
  - Same digital language needed.
  - PED narrative
  - Vision native for cities.
  - Linking PED to city strategies.
  - Better understanding.
  - PED as a marketing tool.
  - Certificate standards (for developer).
  - Only balance not a good marketing.
  - Defining the road towards ped and elements of support.
  - Climate neutrality.
  - Reduced carbon emissions.
  - Tangible outcomes of R&I tools for cities.

- Management
  - Municipalities are complex and implementation is difficulty.
  - Include relevant stakeholders, how to support a PED after completion of the project.
  - Mentioning incentives for stakeholders.
  - Get communities involved.
  - EU regulations as a topic of interest and a national law as a variable in research.
  - Close links between municipalities and partners.
  - Fixed meetings between cities and residents.
  - Personal contracts.
  - Maintenance of cities.
  - Identifying benefits and levels for each layer in a city.

**Iterative training:** these observations and suggestions made by experts highlight the importance of several key factors for linking PEDs to digitalization and circularity.

- These factors include continuity, data, PED narrative, and management. The suggestions emphasize the need for continuous learning, reflection, and exchange between projects to identify common learning lessons, improve data collection and accessibility, and apply digital technologies such as digital twins to shape urban reality.
- The experts also stress the need to develop a clear and compelling narrative for PEDs that is native to cities and linked to their strategies.
- Effective management of PED projects is also essential, and the experts suggest involving relevant stakeholders, communities, and partners in the implementation process, identifying benefits and levels for each layer in a city, and developing personal contracts and fixed meetings between cities and residents to ensure successful implementation and maintenance of cities.

*3.3. Communication and Dissemination of PED*

3.3.1. Project-Related Communication/Public Relations Strategies

*Q: How to communicate Project-related/public relations strategies for PEDs?*

**ChatGPT:** Effective communication is a crucial aspect of implementing PED projects. It is important to create a shared understanding of the project goals and objectives among all stakeholders, including the municipality, residents, local businesses, and energy providers. To communicate project-related strategies, the following approaches can be used:

- Clear and simple messaging: Use clear, concise, and simple language to communicate the project objectives and benefits to all stakeholders.
- Interactive workshops and meetings: Conduct workshops and meetings with stakeholders to gather their inputs and feedback, and to build a sense of community and ownership.
- Use of social media and online platforms: Utilize social media platforms and websites to reach a wider audience and to provide regular updates on the project progress and developments.
- Involvement of local media: Partner with local media outlets to raise awareness about the project and to promote its goals and objectives.
- Visible progress and results: Show visible progress and results to demonstrate the benefits of the project and to encourage continued support and involvement from stakeholders.
- Clearly articulate the goals and benefits of the PED project to stakeholders, including the community, government, and energy providers.
- Use a variety of communication channels, including online platforms, local media, and community events, to reach a wide audience.
- Engage local residents and stakeholders in the communication process by providing opportunities for input and feedback.
- Create clear and easily accessible information materials, such as brochures and videos, that explain the project and its benefits.
- Highlight the successes and achievements of the PED project, and share this information with the community and other stakeholders.
- Foster open and transparent communication between project partners, government entities, and local residents to promote trust and collaboration.
- Continuously evaluate and adapt the communication strategies to ensure they are effective in achieving the goals of the PED project.

**Experts:** The observation and suggestions made by the experts are:

- Building up a repository of information.
- Conduct summer schools.
- Use external channels like social media.
- Use local facilities like labs.

**Iterative training:** these are good suggestions for communicating project-related/public relations strategies for PEDs.

- Building up a repository of information can be useful for providing easy access to project-related materials and resources.
- Conducting summer schools is also a good way to raise awareness and provide training for interested parties.
- Using external channels like social media can be an effective way to reach a wider audience, while using local facilities like labs can help to connect with local communities and stakeholders.

3.3.2. Cooperation with JPI UE on How to (Better) Communicate and Raise Awareness on PED Strategies?

*Q: How to better communicate and raise awareness on PED strategies?*
**ChatGPT:** it may involve the following steps:

- Identifying common goals and objectives: The first step is to align the goals and objectives of both parties and ensure that they are working towards a common goal of promoting PEDs.
- Assessing current communication and awareness activities: Different initiatives and PED stakeholders should conduct a review of their current communication and awareness activities to identify opportunities for collaboration.
- Developing a communication plan: Based on the review, a comprehensive communication plan can be developed to outline the communication goals, target audience, channels, and message content.
- Creating joint communication materials: different initiatives and PED stakeholders can collaborate on creating joint communication materials such as brochures, infographics, and videos to raise awareness and educate the public about PEDs.
- Conducting joint events and webinars: Joint events and webinars can be organized to showcase PEDs and share best practices.
- Measuring and evaluating the impact: Both parties should regularly monitor and evaluate the impact of their communication activities to assess their effectiveness and make improvements as needed.

**Experts:** some observation and suggestions made by the experts are:

- Publications, special issues and newsletter.
- Filter is needed for research reports.
- Popular science.
- Joint events with PED projects and stakeholders.
- Repository.
- Summer schools.
- Use external channels like social media.

**Iterative training:** these observations and suggestions made by the experts about how to better communicate and raise awareness on PED strategies are very helpful.

- It's important to use various channels and methods to reach different audiences and increase the visibility of PED projects.
- Publications, newsletters, and joint events can help raise awareness among stakeholders, while popular science and external channels like social media can help reach a wider audience.
- Building a repository of information can also provide a centralized source of information for interested parties, and summer schools can help build capacity and knowledge among relevant groups.
- A filter for research reports can also help to make the information more accessible and relevant.

### 3.3.3. Link to National and Local Policies and Communities

*Q: How to link PEDs to national & local policies and communities?*

**ChatGPT:** Linking PEDs to national and local policies and communities involves the following steps:

- Awareness raising: Start by educating the public and policy-makers about the benefits and potential of PEDs. This can be done through public events, media campaigns, and stakeholder consultations.
- Policy integration: Align PED goals and objectives with existing national and local energy and sustainability policies. This can be done by working closely with policy-makers and integrating PED initiatives into existing plans and strategies.
- Stakeholder engagement: Involve a range of stakeholders, including local authorities, residents, businesses, and community organizations, in the planning and implementation of PEDs. This can help build support and buy-in for the initiatives and increase their chances of success.
- Community-led approach: Encourage community-led initiatives and initiatives that promote local ownership and involvement. This can help create a sense of ownership and pride in the PEDs and increase their chances of long-term success.
- Partnership building: Foster partnerships between different actors, including local authorities, businesses, and community organizations, to ensure a coordinated and effective approach to PED implementation.

Overall, the success of linking PEDs to national and local policies and communities depends on effective communication, stakeholder engagement, and partnership building.

**Experts:** some bullet points are listed as below

- Use local facilities like labs.
- Clustering centrally through webinars, climate neutral cities mission.
- Structure topics.
- Fundamental issues, lessons learned should be shared with future projects.
- Organize TED-talks.
- Connection and links to climate neutrality.
- Conference series.
- One way communication like websites.
- Information on data for citizens.
- Networking opportunities for projects to share their results.
- Publishing PED failures to improve learning.
- Who to communicate- in the project to prosumers while giving bigger picture.
- Outside the project: city administration.

**Iterative training:** these suggestions and observations made by experts seem to be quite valuable in linking PEDs to national and local policies, as well as engaging communities.

- I agree with the idea of using local facilities like labs and organizing TED-talks to bring the topic closer to the communities.
- Additionally, sharing the fundamental issues and lessons learned from previous projects can help future projects to avoid similar mistakes and be more effective.
- Creating networking opportunities for projects to share their results and publishing PED failures to improve learning can also contribute to better communication and knowledge-sharing.
- It is also essential to consider who to communicate with, such as prosumers within the project and city administrations outside of it.

## 4. Discussion

Through the comparison of the results, we found that: (1) ChatGPT's answers depend on what the model knows already through learning from the existing data source. For some of the questions asked about PEDs, there is no single agreed answer because there are currently diverse human opinions about them. ChatGPT has the potential to mislead

stakeholders by providing them with a seemingly accepted solution. Some of answers from ChatGPT sound plausible but are incorrect or nonsensical because the model lacks practical truth. (2) ChatGPT is sensitive to tweaking in the input phrasing or attempting the same prompt multiple times. For example, given one phrasing of a question, the model claims not to know the answer, but given a slight rephrase, it can answer correctly. (3) The model is often excessively verbose and overuses certain phrases, such as restating that it is a language model trained by OpenAI. Such issues arise from biases in the training data (trainers prefer longer answers that look more comprehensive). (4) If ChatGPT is to be a useful communication device for PEDs, iterative training is necessary. However, ChatGPT can only explain the importance of the feedback or mostly agree with the feedback from authors. It still lacks the ability to critically assess these inputs. (5) ChatGPT is able to produce very general but nevertheless logical explanations, which could be useful for public communications. The answers from ChatGPT somewhat overlap with those of the experts in the workshop. Although the answers are not exactly the same, it is the nature of PEDs with diverse opinions in practice. This may be a barrier for ChatGPT to be widely used for PEDs as it lacks the diverse answers to the context-based questions. On the other hand, those answers from ChatGPT are not so useful for experts and professionals who have already worked with PEDs. ChatGPT needs more training in order to provide useful feedback to the questions at a more advanced level. (6) ChatGPT's answers are often delayed compared to the dynamic updates in reality. ChatGPT relies on the information from the internet, while some updates could be performed a few months before the related reports are available online. For instance, discussions on the term "positive" and its usefulness are currently quite intense. Since generating an actual local energy surplus is almost impossible in many existing neighborhoods, most experts now try to avoid phrasing such as "PED must have an annual energy surplus" and instead have sought to revise the narrative so that it says PEDs should "generate a high share of locally generated renewable energy" or be "actively working towards the generation of an annual energy surplus". (7) Compared to the state-of-the-art chatbots, the ability to generate human-like words and text for ChatGPT is a result of being trained on a diverse range of text from the internet and various sources and OpenAI's RLHF, which allows ChatGPT to generate responses based on the patterns and structures it has learned. The massive amounts of language data from the internet may rapidly change. The ability to access and integrate the online resources, especially when a link is provided in the prompt, into answers will largely improve their authenticity. (8) To be able to assist humans in making real-time decisions and better supporting human meetings, it is necessary for ChatGPT to analyze and generate voice messages [29,30].

In addition to the problems identified here regarding the use of ChatGPT as a communication tool for PEDs, as an AI language model, ChatGPT has other limitations that can affect its ability to produce accurate and meaningful responses. These additional limitations include:

- Lack of common sense: ChatGPT lacks common sense knowledge of PEDs, which can lead to it producing responses that may not be logically consistent or accurate. We found this limitation when we tried to ask similar questions, but ChatGPT responded differently, even with repeated descriptions in one answer. This may be due to its sensitivity to tweaks to the input phrasing.
- Limited contextual understanding: ChatGPT's ability to understand the context of a conversation can be limited, leading to irrelevant or nonsensical responses. For instance, the circumstances surrounding PEDs are different in each context, so there is no clear answer for many of the questions that we asked the model. ChatGPT is not able to understand these differences in PED contexts.
- Inability to learn from experience: ChatGPT cannot learn from experience like humans can, which means it may not improve its responses over time in the same way a human would. As we learnt from the expert's workshops, humans can digest the information via different ways, such as conversation, listening, observation, watching, etc., and

therefore we learn PEDs dynamically, while ChatGPT needs continuous language training, which presents as a limitation.

- Biases: like all AI systems, ChatGPT can be biased based on the data it was trained on. If the training data are biased, ChatGPT's responses may reflect those biases. This is a common limitation of all AI tools, which can bring biases when disseminating PEDs.
- Lack of emotional intelligence: ChatGPT may have difficulty recognizing and responding appropriately to the emotional nuances behind a written transcript of a verbal conversation, which can impact its ability to provide empathetic or sensitive responses. During the experts' workshop, there were debates about several questions, and some answers from experts were rather euphemistic. ChatGPT cannot yet fully identify these aspects.
- Inability to reason: ChatGPT is not able to reason through complex or abstract concepts, which can limit its ability to provide insightful or nuanced responses. This limitation could bring forward usefulness when using ChatGPT disseminates PEDs to non-professional stakeholders.
- Lack of creativity: while ChatGPT can generate novel responses, it may not be able to generate truly creative or innovative responses that go beyond what it was trained on. However, we see that this limitation will not substantially influence dissemination of PEDs.

Apart from these functional limitations, the use of ChatGPT to study complex concepts, such as PEDs, could raise several ethical and social implications. These are, amongst others:

- Data privacy issues: using ChatGPT to study complex concepts requires large amounts of data, which may include personal or sensitive information. Ensuring that these data are collected and used ethically, with the consent of those whose data is being used, is important to protect privacy and prevent potential harm.
- Bias issues: as an AI model, ChatGPT is only as unbiased as the data on which it is trained. Therefore, it is important to ensure that the model is built on representative training data. Researchers should be aware of the potential for the model to generate biased responses, especially if it is used to analyze social or political issues, such as those which are relevant in discussions around PEDs.
- Misinformation issues: ChatGPT may generate responses that are inaccurate or misleading, which could have potential negative consequences if the information is used to inform decision-making or policy. Researchers should be careful to evaluate the accuracy of the model's responses and take steps to mitigate any potential harm caused by misinformation. A filter should be put in place when dealing with a complex concept such as PEDs.
- Lack of human oversight: while ChatGPT can be trained to analyze complex concepts, it may not be able to reason or provide insights beyond its training data. Therefore, relying solely on the model to analyze complex concepts without human oversight could lead to inaccurate or incomplete conclusions.
- Responsibility issues: as with any AI model, researchers using ChatGPT to study complex concepts have a responsibility to ensure that the model is used ethically and that any potential harm caused by the model is minimized.
- Impact on employment: the use of ChatGPT to study complex concepts could potentially replace human researchers, which could have negative impacts on employment and the economy. It is important to consider the social implications of AI-based research and ensure that any potential negative impacts are addressed.

## 5. Conclusions

The implementation of PEDs has a significant impact on buildings and requires approaches encompassing building design, construction practices, the development of new building types, and the establishment of regulatory frameworks so that it can create sustainable, energy-efficient, and environmentally friendly built environments. PEDs promote the integration of renewable energy sources and energy-efficient technologies into sustainable

buildings. This requires changes in the way buildings are designed and constructed to incorporate energy generation systems. PEDs often encourage the development of new building types that prioritize energy efficiency, renewable energy generation, and smart grid integration. To support these initiatives, governments and local authorities could update building codes to include energy performance standards, renewable energy requirements, and incentives for sustainable building. Thus, the use of ChatGPT will also generate an impact on buildings.

This study not only validated the possibility of using ChatGPT to fast study PEDs but also trained ChatGPT by feeding back some of the experts' opinions into the tool, which is expected to find a way on fast information dissemination through ChatGPT.

ChatGPT could be a useful tool for the communication and dissemination of information and ideas about PEDs. However, a few specific limitations have been observed. ChatGPT is sensitive to the phrasing of questions, is prone to repeating non-important (or general) information, lacks the ability to critically assess inputs, is unable to explore online resources, and is not able to provide diverse answers to context-based questions. Its answers are mostly drawn from the internet and annotated by a limited number of human evaluators, which cannot reflect the diversity and distinctiveness of verbal human interactions. These limitations constrain its usefulness in some scenarios.

It is believed that ChatGPT could play a limited role in the rapid transference of information and knowledge of PEDs to the public. Additional research is required to explore the solutions that could be implemented to overcome ChatGPT's limitations, such as different forms of iterative training. Involving ChatGPT in the live PED workshop sessions may have been one way of doing this, but this will be more efficient when it has additional voice function similar to a real chatbot.

Moreover, while ChatGPT is an impressive AI language model, it is important for the general public to recognize its common limitations and understand that it may not always produce accurate or meaningful responses regarding PEDs. On the other hand, while the potential of ChatGPT to offer rapid formulations of complex concepts, such as PEDs, is considerable and has much to contribute to advances in research, it is important to consider the ethical and social implications of using such models and take steps to mitigate any potential harm.

**Author Contributions:** Conceptualization, X.Z.; methodology, X.Z., J.S. and M.H.; formal analysis, X.Z., J.S. and M.H; writing—original draft preparation, X.Z., J.S. and M.H.; writing—review and editing, X.Z., J.S. and M.H. All authors have read and agreed to the published version of the manuscript.

**Funding:** This project receives funding under the Joint Programming Initiative (JPI) Urban Europe framework. It receives funding support from the strategic innovation program "Viable Cities", which is financed by Vinnova, the Swedish Energy Agency and Formas (P2022-01000), The Scientific and Technological Research Center of Turkey (Türkiye), and Austrian Federal Ministry for Climate Action, Environment, Energy, Mobility, Innovation, and Technology (BMK). This research was funded by the Swedish Energy Agency with grant number 8569501.

**Data Availability Statement:** Not applicable.

**Acknowledgments:** We would like to thank all the experts and acknowledge their contributions to this paper at the workshops on 30–31 January 2023 in Copenhagen, Demark. The authors also acknowledge that ChatGPT was used to generate content in this paper. We also thank Janice Holmes from Dalarna University for improving the English grammar in this paper.

**Conflicts of Interest:** The authors declare no conflict of interest.

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
