# Peer review of "ChatGPT for Fast Learning of Positive Energy District (PED): A Trial Testing and Comparison with Expert Discussion Results"

_buildings, doi:10.3390/buildings13061392_

Round 1
Reviewer 1 Report (New Reviewer)
1- Abstract
The abstract effectively summarizes the purpose, methodology, findings, limitations, and recommendations of the study.
It provides a clear understanding of the research context, the role of ChatGPT, and its potential contributions and limitations.
The abstract could benefit from more specific information about the trial test and the key findings derived from the comparison between ChatGPT formulations and expert-derived formulations.
Including a brief mention of the methodology used to train ChatGPT could provide more insight into the model's capabilities.
Overall, the abstract effectively communicates the main points of the paper and generates interest in the study.
2- Introduction
The introduction provides a comprehensive overview of PEDs, their significance, and the challenges associated with their design, planning, implementation, and operation.
It effectively establishes the context and importance of PEDs in achieving sustainability goals.
The mention of existing studies adds credibility to the topic, but specific references or citations would further enhance the introduction.
The introduction could benefit from a clearer structure to better distinguish between different stages (design, planning, implementation, and operation) and their associated challenges.
Overall, the introduction provides a solid foundation for the discussion of PEDs and sets the stage for further exploration in the subsequent sections.
3- References
Enrich your work by citing recent published work presented the power of ChatGPT suc as
https://journals.mesopotamian.press/index.php/CyberSecurity/article/view/33
https://journals.mesopotamian.press/index.php/CyberSecurity/article/view/29
4- Conclusion
The conclusion effectively summarizes the findings regarding the utility of ChatGPT in studying and disseminating information about PEDs.
It provides a clear understanding of the limitations of ChatGPT and the potential challenges that need to be addressed for its optimal use.
The mention of involving ChatGPT in live workshops and considering additional functionalities, such as voice capability, shows a forward-thinking approach to improving its usefulness.
The emphasis on public awareness and the ethical considerations surrounding the use of AI models like ChatGPT is important and demonstrates a responsible perspective.
Overall, the conclusion provides a thoughtful reflection on the findings and opens up avenues for future research and improvement in utilizing ChatGPT for PED-related information dissemination.
none
Author Response
Please see the attachment.

Reviewer 2 Report (New Reviewer)
Need for improvement mainly on References and Conclusion

Author Response
Please see the attachment.

This manuscript is a resubmission of an earlier submission. The following is a list of the peer review reports and author responses from that submission.
Round 1
Reviewer 1 Report
The abstract required improvement for clarity in understanding
About NLP has to be included
English grammar to be improved a lot for the entire article
improve the number of literature survey
Author Response
Dear editor,
The detailed response to the reviewer's comments is attached in the file.
Best regards
The authors

Reviewer 2 Report
The manuscript is still weak, only a few references have been added, which worries me since other tools could be compared, more authors could be investigated to know what they think of these tools and to make a good discussion. The conclusions are not well done because their references are not enough, this manuscript could be of great interest but it is not enough in the way it is presented.
I did a quick search in Scopus and found 5,739 papers on open ai.
Author Response

(The authors gave the same response as above.)

Reviewer 3 Report
The paper is recommended for acceptance.
Author Response

(The authors gave the same response as above.)

Round 2
Reviewer 2 Report
My decision is still to reject the manuscript, not knowing the identity of the "experts" creates a lot of uncertainty, so much so that one might think that some parts of the manuscript has been written by the GTP chat itself.